# Synthesis and Evaluation of Reactive Oxygen Species Sensitive Prodrugs of a NAMPT Inhibitor FK866

**DOI:** 10.3390/molecules28010169

**Published:** 2022-12-25

**Authors:** Zili Xu, Huihui Wang, Haixia Liu, Hongli Chen, Biao Jiang

**Affiliations:** 1Shanghai Institute for Advanced Immunochemical Studies, ShanghaiTech University, 393 Middle Huaxia Road, Pudong, Shanghai 201210, China; 2School of Physical Science and Technology, ShanghaiTech University, 393 Middle Huaxia Road, Shanghai 201210, China; 3University of Chinese Academy of Sciences, 19A Yuquan Road, Shijingshan District, Beijing 100049, China; 4School of Life Science and Technology, ShanghaiTech University, 393 Middle Huaxia Road, Shanghai 201210, China

**Keywords:** NAMPT inhibitor, FK866, ROS-responsive release, prodrug

## Abstract

NAMPT is an attractive target in cancer therapy and numerous NAMPT inhibitors have been developed. However, the clinical activities of NAMPT inhibitors have displayed disappointing results in clinical trials for their dose-limiting toxicities. In this study, reactive oxygen species (ROS)-responsive prodrugs of a NAMPT inhibitor FK866 were designed and synthesized. A short synthesis method was developed to shield the activity of FK866 through a quaternary ammonium connection. Two prodrugs, with boronic acid as a responsive group to ROS, were prepared and one of the prodrugs **122-066** also contained a fluorescence carrier. Both of the prodrugs released the active compound by the treatment of H_2_O_2,_, and the biological evaluation showed that they exhibited a higher potency in cells with high levels of ROS. Moreover, prodrug **122-066** had the ability to release FK866 and simultaneously induce the fluorescence activation under the stimulation of H_2_O_2_. This method has the potential to improve the therapeutic window of NAMPT inhibitors.

## 1. Introduction

Nicotinamide phosphoribosyltransferase (NAMPT), a rate-limiting enzyme, is crucial for nicotinamide adenine dinucleotide (NAD^+^) biosynthesis which is involved in a broad range of biological processes [1,2]. NAMPT has drawn considerable interest in the fields of metabolism, senescence, immune response, and cancer [3,4]. Emerging evidences have shown that NAMPT is overexpressed in a broad range of tumor cells, and it holds great promise to identify NAMPT inhibitors as potential therapies for cancers [5,6]. The search for novel anti-tumor candidates has led to the development of numerous NAMPT inhibitors, and two compounds FK866 and CHS-828 have entered clinical trials [7,8,9]. FK866 has been extensively studied in preclinical research and shows robust antineoplastic potency across various solid and hematological cancers [8,10,11]. Unfortunately, the clinical activity of FK866 has shown disappointing results in Phase II trials due to its dose limiting toxicities [8]. The failure of FK866 and other NAMPT inhibitors in clinical trials has motivated research endeavors to explore new strategies to widen the therapeutic window, and maximize therapeutic efficacies for NAMPT inhibitors in oncology. Proteolysis-targeting chimera (PROTAC) technology has been used to prepare a NAMPT degrader (Figure 1a) [12,13]. NAMPT inhibitors have also been developed as payloads of antibody-drug conjugates (ADCs), which can selectively deliver NAMPT inhibitors to the targeted cancer cells. (Figure 1b) [14,15,16] Methods to enhance the therapeutic opportunities of NAMPT inhibitors remain attractive and require attention.

A prodrug is a well-established approach to improve drug properties and increase efficacy. Active drugs link to temporary moieties to form prodrugs, which are designed to be activated after administration by chemical or enzymatic reactions [17]. Recently, the overexpressed levels of reactive oxygen species (ROS) have gained considerable attention because of their crucial role in a variety of diseases, especially cancer [18]. Significant differences in ROS levels have been observed between the normal cells and cancer cells [19]. Generally, the endogenous ROS levels are maintained at a range of 20 × 10^−3^ μM in healthy cells. Whereas, ROS levels in cancer cells can reach high concentrations of up to 100 μM [20,21]. Thus, ROS have been explored as a trigger to promote prodrug release of the parent drug, and have been successfully applied in the selective delivery of drugs and precise diagnosis of ROS-related diseases [18,22].

Although ROS-responsive prodrugs are well defined and are proven to be efficient in reducing the toxicities of the parent compound, these approaches have not been used in the preparation of NAMPT-related inhibitors. Indeed, the strategy of prodrugs has been employed for NAMPT inhibitor CHS-828 by the introduction of a PEG chain (EB162, also named GMX1777) to improve its pharmacokinetic and solubility profile (Figure 1c) [23]. However, this method has not alleviated the side effects of its parent drug, and the prodrug has also failed in the clinical stage [9]. To reduce undesired toxicities of NAMPT inhibitors, in this study we aim to develop ROS-responsive prodrugs of FK866. We envision that these ROS-activated NAMPT inhibitors would target cancer cells with high levels of ROS. Herein, prodrugs of FK866 were designed, synthesized, and their biological activities were evaluated. 

## 2. Results and Discussion

Crystallographic studies of the NAMPT-FK866 complex showed that FK866 binds within the enzyme catalytic domain, and the pyridyl ring of FK866 is essential to its binding activities (Figure 2) [6]. Benefiting from the structure information, we assumed that modifying the pyridyl ring could significantly block the activity of FK866. Therefore, a short synthesis approach was designed to yield compound **121-001** that was supposed to shield the biological activities of FK866. Specifically, compound **121-001** was prepared by the attachment of a benzyl group to the pyridyl ring of FK866 through a quaternary ammonium salt connection (Figure 1). 

Boronic acid (and ester) has been well developed as a promising kind of chemical functional group to respond to ROS [24]. The boron atom is susceptible to be attacked by ROS members, such as H_2_O_2_, to produce a hydroxyl group, which may trigger bond breaking and subsequently release the active compound [18]. Similarly, through a quaternary ammonium formation, the commercially available compound 4-(Bromomethyl)benzeneboronic acid pinacol ester was incorporated into FK866, producing the boronic ester **1**, which was further hydrolyzed to obtain the boronic acid **121-052**. Moreover, a prodrug with a fluorescence carrier was also developed [25,26]. Coumarin skeleton **2**, that was used as a fluorophore, was prepared by the reaction of 4-bromo-2-hydroxybenzaldehyde with propionic anhydride. Then, compound **2** was reacted with bis(pinacolato)diboron to yield the boronate derivative **3**. Bromination of the benzylic position of compound **3** occurred to give the intermediate **4**, which was linked to FK866, yielding compound **5**. Finally, the desired prodrug **122-066** was obtained by removal of the pinacol ester (Figure 1).

The stability of the prodrugs **122-052** and **122-066** was evaluated in PBS (Phosphate Buffered Saline, pH 7.4) and the liver microsome. The results showed that both of the compounds are very stable in the PBS, and only a tiny compound degraded during the incubation for four days. In the liver microsome, **122-052** still remained with high stability, while **122-066** degraded with time (Figure 3). 

To evaluate whether prodrugs **122-052** and **122-066** were able to release the parent compound FK866, their reactivity was examined in the presence of H_2_O_2_, and monitored by high-performance liquid chromatography (HPLC). As shown in Figure 4a, when compound **122-052** was treated with 100 μM H_2_O_2_, the oxidative intermediate **6** appeared quickly and the active compound FK866 was also detected immediately. Subsequently, compound **6** further converted completely to FK866 with the extension of time. As shown in Figure 4b, compound **122-066** was also treated with 100 μM H_2_O_2_. Similarly, the phenol intermediate **7** was observed firstly, and then the released FK866 appeared. However, the conversion process took a lot longer. Benefiting the coumarin fluorophore, the chemical conversion of compound **122-066** under the treatment of H_2_O_2_ was also detected by fluorescence spectroscopy. The boronate moiety was oxidated by H_2_O_2_, and following a 1,6-elimination reaction, FK866 and the coumarin probe **8** were released. Probe **8** would then convert into compound **9** upon reaction with H_2_O. (Figure 4c). As shown in Figure 4d, prodrug **122-066** displayed weak fluorescence at 450 nm, and the fluorescence appeared in the presence of H_2_O_2_. When the concentration of H_2_O_2_ was increased, which would accelerate the rate of the reaction to release the fluorescent probe, the increase in fluorescence was observed.

Based on the positive results that H_2_O_2_ mediated the release of FK866, we next investigated the biological activities of FK866 and the synthesized compounds **122-001**, **122-052** and **122-066** on different cell lines. Accumulating evidences have shown that cancer cells yield elevated levels of ROS. Firstly, cell lines, including 293T (human embryonic kidney), Molt 4 (human T lymphoblast; acute lymphoblastic leukemia), and PC-3 (human prostate cancer cells) were chosen, and their ROS levels were detected by FACS (Fluorescence Activated Cell Sorting). DCFH-DA (2,7-Dichlorodihydrofluorescein diacetate) was used as an indicator for ROS in cells. The results showed that the ROS level in PC-3 was much higher than in 293T and Molt 4 cells (Figure 5a). Then, we evaluated the anti-proliferative activities of the compounds against the above mentioned cells. As expected, FK866 exhibited high potent anti-proliferative activity against all the cells. The potency of compound **122-001** decreased dramatically and it showed negligible inhibition activity for all the cells, confirming its capability to shield the activity of the parent compound. Whereas for prodrugs **122-052** and **122-066**, they were more sensitive to the cells with high ROS level and significant differences were observed for PC-3 cells, compared with 293T and Molt 4 cells (Figure 5b).

To provide further confirmation for H_2_O_2_-mediated release of the active compound, along with the fluorescence activation for the prodrug **122-066**, cellular fluorescence images were performed with the PC-3 cell line. A weak fluorescence signal appeared in the absence of H_2_O_2_ (Figure 6, lane A). When cells were treated with prodrug **122-066** (0.93 nM) for 6 h, and incubated for another 12 h after the addition of H_2_O_2_ (100 μM), a significant fluorescence signal was observed (Figure 6, lane A). The result confirmed that H_2_O_2_ triggered the release of the active compound, and induced he fluorescence activation for prodrug **122-066**. The cellular uptake capability of prodrug **122-066** was also investigated. Cells were treated with **122-066** for 6 h (Figure 6, lane C) or 12 h (Figure 6, lane D), and then **122-066** was removed by washing with PBS. A further 100 μM H_2_O_2_ was added, and the cells were incubated for another 12 h. Fluorescence signals appeared for both groups and an enhanced fluorescence intensity was observed for the group that was treated with **122-066** for a longer period, indicating the cellular delivery ability of **122-066**.

Cell viability test on native and activated PBMC (peripheral blood mononuclear cell) by phytohematoagglutinin (PHA, 5 µg/mL) was performed. The results showed that FK866 inhibited the cell viability remarkably on both native and activated PBMC at 4 nM. Whereas, cells were mostly unaffected by the treatment of prodrugs **122-052** and **122-066** at the same concentration (Figure 7).

## 3. Materials and Methods

### 3.1. Chemistry

All chemical reagents were of analytical grade, obtained from commercial sources, and used as supplied without further purification unless indicated.

NMR spectra were recorded on a Bruker-500 (500 MHz) instrument (Appendix A). The deuterated solvents employed were purchased from Energy Chemical. Chemical shifts were given in ppm with respect to referenced solvent peaks. Spectra were analyzed with MestReNova. High-resolution mass spectra (HRMS-ESI) were obtained on an ABsciex 4600. Analytical high performance liquid chromatography (HPLC) was performed on SHIMADZU LC-30 AD machine, using an Agela Technologies C18 column (2.1 × 100 mm, 3 µm). LC system: solvent A: 0.5% (*v*/*v*) TFA in H_2_O; solvent B: acetonitrile at 30 °C; gradient: 0–10 min 10–100% B, 10-12 min 100% B at flow rate of 0.4 mL/min; detector: UV detection (λmax = 220–254 nm). Fluorescence intensity and emission spectrum was read by microplate reader. 

FK866 was synthesized as described by Galli et al [27].

The general procedure for synthesis of pyridinium boronic acid (**122-052**, **122-066**): To the solution of FK866 in anhydrous MeCN, 1 equivalent of NaHCO_3_, and 1.2 equivalents of different bromides were added. This reaction mixture was heated under 55 °C for 12 h. The solvent was removed under reduced pressure to obtain the crude mixture of boronic acid and its pinacol ester. The resulting residue was re-dissolved by MeOH (3 mL), followed by addition of HCl (1 M, 1 mL). This mixture was stirred for 3 h and purified by HPLC (using 10 to 100% MeCN in 0.5% HCl).

(*E*)-3-(3-((4-(1-benzoylpiperidin-4-yl)butyl)amino)-3-oxoprop-1-en-1-yl)-1-(4-boronobenzyl)pyridin-1-ium (**122-052**) (yield = 42%): ESI-HRMS calcd for C31H37BN3O4 [M^+^]: 526.2872 found: 526.2914. ^1^H NMR (500 MHz, MeOD) δ 9.39 (s, 1H), 8.99 (d, *J* = 6.0 Hz, 1H), 8.79 (d, *J* = 8.1 Hz, 1H), 8.11 (dd, *J* = 8.2, 6.0 Hz, 1H), 7.81 (d, *J* = 7.6 Hz, 2H), 7.60 (d, *J* = 15.7 Hz, 1H), 7.50 (d, *J* = 7.8 Hz, 2H), 7.49–7.41 (m, 3H), 7.41–7.34 (m, 2H), 7.02 (d, *J* = 15.8 Hz, 1H), 5.88 (s, 2H), 4.60 (d, *J* = 12.7 Hz, 1H), 3.69 (d, *J* = 13.4 Hz, 1H), 3.36–3.32 (m, 2H), 3.08 (t, *J* = 12.8 Hz, 1H), 2.83 (t, *J* = 12.3 Hz, 1H), 1.84 (d, *J* = 13.2 Hz, 1H), 1.68 (d, *J* = 13.3 Hz, 1H), 1.58 (pent, *J* = 7.0 Hz, 3H), 1.46–1.28 (m, 4H), 1.23–1.18 (m, 1H), 1.14–1.05 (m, 1H). ^13^C NMR (126 MHz, MeOD) δ 170.94, 164.95, 143.98, 143.87, 143.00, 136.52, 135.59, 134.63, 131.90, 129.67, 128.72, 128.38, 128.31, 127.78, 126.40, 64.50, 48.50, 48.09, 42.49, 39.30, 35.67, 35.63, 32.50, 31.69, 29.08, 23.63.

(*E*)-3-(3-((4-(1-benzoylpiperidin-4-yl)butyl)amino)-3-oxoprop-1-en-1-yl)-1-((7-borono-2-oxo-2H-chromen-3-yl)methyl)pyridin-1-ium (**122-066**) (yield = 49%): ESI-HRMS calcd for C34H37BN3O6 [M^+^]: 594.2770, found: 594.2819. ^1^H NMR (500 MHz, MeOD) δ 9.38 (d, *J* = 1.6 Hz, 1H), 9.10 (d, *J* = 6.1 Hz, 1H), 8.78 (dt, *J* = 8.2, 1.4 Hz, 1H), 8.45 (s, 1H), 8.11 (dd, *J* = 8.2, 6.1 Hz, 1H), 7.73 (s, 2H), 7.61 (d, *J* = 15.8 Hz, 1H), 7.47–7.42 (m, 3H), 7.37 (dh, *J* = 4.7, 2.6 Hz, 2H), 6.97 (d, *J* = 15.8 Hz, 1H), 5.77 (s, 2H), 4.60 (d, *J* = 13.0 Hz, 1H), 3.69 (d, 1H), 3.33 (d, *J* = 7.0 Hz, 1H), 3.07 (t, 1H), 2.82 (t, 1H), 1.84 (d, 1H), 1.70–1.64 (m, 1H), 1.61–1.54 (m, 3H), 1.46–1.36 (m, 2H), 1.36–1.30 (m, 2H), 1.23–1.08 (m, 1H). ^13^C NMR (126 MHz, MeOD) δ 170.94, 164.95, 160.94, 153.57, 145.92, 144.80, 144.51, 143.09, 136.09, 135.94, 131.99, 129.52, 128.48, 128.31, 127.91, 127.73, 126.35, 60.87, 56.25, 56.08, 55.91, 47.90, 42.31, 39.27, 35.73, 35.65, 32.54, 31.68, 29.08, 23.59, 16.21, 16.05, 15.90, 15.75.

(*E*)-3-(3-((4-(1-benzoylpiperidin-4-yl)butyl)amino)-3-oxoprop-1-en-1-yl)-1-benzylpyridin-1-ium (**122-001**) (yield = 77%): To the solution of FK866 (20 mg, 0.051 mmol) in anhydrous MeCN, NaHCO_3_ (5.12 mg, 0.061 mmol), and 4-(Bromomethyl)benzeneboronic acid pinacol ester (14.5 mg, 0.061 mmol) were added. The reaction mixture was heated under 55 °C for 12 h. The solvent was removed under reduced pressure. The resulting residue was re-dissolved by MeOH and filtered, followed by purification using HPLC (using 10 to 100% MeCN in 0.5% HCl) (yield = 64%) ESI-HRMS calcd for C31H36N3O2 [M^+^]: 482.2802 found: 482.2823. ^1^H NMR (500 MHz, MeOD) δ 9.35 (s, 1H), 8.98 (d, *J* = 5.9 Hz, 1H), 8.79 (d, *J* = 8.0 Hz, 1H), 8.14–8.08 (m, 1H), 7.61 (d, *J* = 15.8 Hz, 1H), 7.56–7.53 (m, 2H), 7.50–7.43 (m, 4H), 7.37 (dq, *J* = 6.8, 2.6 Hz, 2H), 6.99 (d, *J* = 15.8 Hz, 1H), 5.87 (s, 2H), 4.60 (d, *J* = 12.8 Hz, 1H), 3.69 (d, *J* = 13.5 Hz, 1H), 3.34 (d, *J* = 7.1 Hz, 1H), 3.08 (t, *J* = 12.9 Hz, 1H), 2.83 (t, *J* = 12.7 Hz, 1H), 1.85 (d, *J* = 13.3 Hz, 1H), 1.68 (d, *J* = 13.2 Hz, 1H), 1.59 (pent, *J* = 7.1 Hz, 3H), 1.46–1.30 (m, 4H), 1.23–1.09 (m, 1H). ^13^C NMR (126 MHz, MeOD) δ 170.95, 164.94, 142.89, 136.57, 133.07, 131.89, 129.74, 129.52, 129.37, 128.78, 128.71, 128.31, 126.35, 64.61, 39.30, 35.73, 35.65, 29.09, 23.61.

7-bromo-3-methyl-2H-chromen-2-one (**2**): To the mixture of 4-Bromo-2-hydroxybenzaldehyde (1005 mg, 5 mmol) and sodium propionate (960.6 mg, 10 mmol), propionic anhydride (960.6 mg, 15 mmol) and TEA (506 mg, 5 mmol) were added. The reaction mixture was stirred overnight under 170 °C. The resulting mixture was diluted with EA and washed by saturated NaHCO_3_ aqueous solution and brine. The organic phase was collected, dried by Na_2_SO_4_, and purified by flash column chromatography (using 10 to 30% EA in PE). (yield=34%) ESI-HRMS calcd for C_10_H8BrO2 [(M + H)^+^]: 238.9708 found: 238.9718. ^1^H NMR (500 MHz, CDCl3) δ 7.50–7.45 (m, 2H), 7.38 (dd, *J* = 8.2, 1.8 Hz, 1H), 7.27 (s, 1H), 2.20 (d, *J* = 1.3 Hz, 3H). ^13^C NMR (126 MHz, MeOD) δ 170.95, 164.94, 142.89, 136.57, 133.07, 131.89, 129.74, 129.52, 129.37, 128.78, 128.71, 128.31, 126.35, 64.61, 39.30, 35.73, 35.65, 29.09, 23.61.

3-methyl-7-(4,4,5,5-tetramethyl-1,3,2-dioxaborolan-2-yl)-2H-chromen-2-one (**3**): To the solution of compound 3 (316 mg, 1.33 mmol) in dried 1,6-dioxane, bis(pinacolate)diborone (370.75 mg, 1.46 mmol), AcOK (391.62 mg, 3.99 mmol) and Pd(dppf)Cl_2_ (48.7 mg, 0.067 mmol) were added and stirred overnight under 85 °C in Argon atmosphere. The solvent was removed under reduced pressure. The resulting residue was re-dissolved by DCM and purified by flash column chromatography (using 10 to 20% EA in PE) (yield = 74%). ESI-HRMS calcd for C16H20BO4 [(M + H)^+^]: 287.1455, found: 287.1466. ^1^H NMR (500 MHz, CDCl3) δ 7.71 (s, 1H), 7.65 (dd, *J* = 7.6, 1.0 Hz, 1H), 7.51 (t, *J* = 1.6 Hz, 1H), 7.40 (d, *J* = 7.6 Hz, 1H), 2.22 (d, *J* = 1.4 Hz, 3H), 1.36 (s, 11H), 1.26 (s, 3H). ^13^C NMR (126 MHz, CDCl3) δ 162.21, 152.67, 138.99, 130.13, 127.01, 126.17, 122.38, 121.67, 84.32, 83.50, 25.03, 24.87, 17.33.

3-(bromomethyl)-7-(4,4,5,5-tetramethyl-1,3,2-dioxaborolan-2-yl)-2H-chromen-2-one (**4**): To the solution of compound **3** (61 mg, 0.21 mmol) in CCl_4_ (2 mL), N-bromosuccinimide (53 mg, 0.30 mmol) and AIBN (2 mg cat.) were added. This reaction mixture was stirred overnight under 85 °C. The mixture was purified by flash column chromatography (using 100% DCM) (yield = 51%). ESI-HRMS calcd for C16H18BBrO4 [(M + H)^+^]: 365.0560, found: 365.0557 and 367.0533. ^1^H NMR (500 MHz, CDCl_3_) δ 7.86 (s, 1H), 7.75 (s, 1H), 7.70 (dd, *J* = 7.6, 1.0 Hz, 1H), 7.49 (d, *J* = 7.7 Hz, 1H), 4.44 (d, *J* = 0.8 Hz, 2H), 1.37 (s, 12H). ^13^C NMR (126 MHz, CDCl_3_) δ 159.96, 153.12, 141.72, 130.51, 127.22, 126.45, 122.61, 120.87, 84.52, 77.27, 29.72, 27.62, 24.89. 

### 3.2. HPLC Analysis of Drug Release Triggered by H_2_O_2_

Prodrugs **122-052**, **122-066** were dissolved in DMSO to prepare a 10 mM stock separately. This solution was diluted to 100 μM in PBS (pH = 7.4) and pre-heated to 37 °C. To this solution, H_2_O_2_ (10 mM) was added to set its final concentration to 100 μM. This reaction mixture was vibrated under 37 °C. Samples were taken at different time points, and directly analyzed by HPLC without further dilution.

### 3.3. HPLC Analysis of Stability of Prodrugs

Prodrugs **122-052**, **122-066** were dissolved in DMSO to prepare a 10 mM stock separately. This solution was diluted to 100 μM in PBS (pH = 7.4) and pre-heated to 37 °C. To 200 μL of this solution, rat liver microsome (5 μL, Gibico, New York, NY, USA) was added, followed by NADPH (10 μL 20 mM in H_2_O). This mixture was vibrated under 37 °C. Samples were taken at different time points (20 μL). To those samples with protein, 40 μL of MeCN was added and the samples were centrifugated (15 min 15000 rpm). The supernatant was analyzed by HPLC.

### 3.4. Fluorescence Assay

To the solution of prodrug **122-052**, **122-066** in PBS (100 μL, 1 μM, pH = 7.4), 1 μL of a different concentration of H_2_O_2_ was added. The excitation was carried out at 365 nm, and the detection of the emission spectrum was set between 400 nm and 500 nm, step by 2 nm.

### 3.5. Cell lines and Culture 

Cell lines 293T, Molt-4, and PC3 were purchased from American Type Culture Collection (ATCC, Manassas, VA, USA). The 293T complete medium was DMEM (Meilunbio, Dalian, China) supplemented with 10% Fetal bovine serum (Sunrise, Claymont, DE, USA) and 1% Penicillin-Streptomycin (Meilunbio, Dalian, China); the complete medium for PC3 and Molt-4 cells was RPMI 1640 (Meilunbio, Dalian, China) supplemented with 10% Fetal bovine serum and 1% penicillin-streptomycin. All cells were incubated aseptically in a 37 °C incubator containing 5% CO_2_.

### 3.6. Reactive Oxygen Species ASSAY

Adequate amounts of 293T, Molt-4 and PC3 were collected in 1.5mL centrifuge tubes. DCFH-DA (Meilunbio, Dalian, China) was diluted with serum-free medium at 1:1000 dilution to achieve a final concentration of 10 μM. The cells collected in the 1.5 mL centrifuge tube were resuspended by the diluted DCFH-DA working liquid to a density of 1.5*10E6 cells/mL and were incubated for 30 min without light. After washing with PBS 2 times, the fluorescence intensity was detected by CytoFLEX flow cytometer (Beckman Coulter, Brea, CA, USA), and the data were processed by FlowJo 7.

### 3.7. In Vitro Cytotoxicity Assays

The starting point of the small molecule drug was 100 nM, and 5-time dilutions were performed successively, with 8 concentration points in total. Suspended cells, Molt-4, were laid on 96-well plate with 20,000 cells/well, after treatment with diluted drugs in the incubator for 48 h. Adherent cells 293T (3000 cells/well) and PC3 (4000 cells/well) were placed on 96-well plates, cultured overnight and were added with diluted drugs in the incubator for 72 h. All cells incubated with the drug were tested for cell viability using the Cell Counting Kit-8 (CCK-8) kit (Meilunbio, Dalian, China).

### 3.8. Viability Assay

PBMC were seeded at 50,000 per well in 96-well plates. Proliferation was induced by adding PHA (5 µg/mL). After 72 h of small molecule drug treatment, cell viability was tested by CellTiter-Glo^®^ Luminescent Cell Viability Assay (Promega, Beijing, China).

### 3.9. Fluorescence Imaging

According to the experimental design, the compounds were added to 6-well plates and incubated for different times; H_2_O_2_ was added to the plates for 12 h. Cells were fixed with 4% PFA at room temperature for 15 min, and washed twice with PBS. Cells were penetrated with 0.1% PBST (0.1% Triton X-100 in PBS) for 30 min at room temperature. Cells were incubated with PI for 20 min at room temperature, washed once with PBS, and sealed with a mounting solution. Images were obtained with a laser confocal microscope (LSM 710).

## 4. Conclusions

NAMPT is an attractive target in cancer therapy, and the development of NAMPT inhibitors represents a promising therapeutic approach. However, the toxicity of NAMPT inhibitors in clinical trials have hindered the development of such drugs. Prodrugs triggered by the tumor microenvironment represent a promising area for the development of selective anticancer chemotherapy with a wider therapy window. In this study, ROS-responsive prodrugs of NAMPT inhibitor FK866 were designed and synthesized. A short synthesis approach was developed to shield the activity of FK866 through the quaternary ammonium connection. The prodrugs released the active compound under ROS conditions. The biological activities of the prodrugs showed that they exhibited higher potencies in cells with high levels of ROS. Meanwhile, the toxicity of the prodrugs 122-052 and 122-066 was significantly reduced on both native and active PBMCs, compared with FK866. Moreover, a fluorescent signal was observed when the cells were treated with H_2_O_2_ and prodrug 122-066 that has a fluorescence carrier indicating its ability for cellular delivery. However, the coumarin moiety can cause the instability of prodrug 122-066 in rat liver microsomes, which is probably because coumarin can be metabolized by liver microsomes. [28,29] As far as we know, this is the first report of a ROS-activated NAMPT prodrug, which offers a practical application and experimental validation of the methods used to control NAMPT inhibitors’ activation. We envision this strategy will aid in and inspire future research for NAMPT-targeted inhibitors in oncology, with maximized therapeutic opportunities.

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
