# Peer review of "Synthesis and Evaluation of Reactive Oxygen Species Sensitive Prodrugs of a NAMPT Inhibitor FK866"

_molecules, 2022, doi:10.3390/molecules28010169_

Round 1
Reviewer 1 Report
This manuscript by Jiang et al. describes antitumor prodrugs of NAMPT inhibitors masked by a benzeneboronate moiety activatable by ROS in vitro. The technology relies on the boron-carbon bond oxidation by ROS such as hydrogen peroxide and further quinone methide self-immolation process for the release of the inhibitor FK866.
Overall, the chemistry and the biological results are sound and this study represents an interesting contribution. I would like to support the publication in Molecules provided that the authors answer my questions and remarks below.
- The comparative assay of ROS production by different cell lines led to the identification of PC-3 as the most productive cell line. The probe DCFH-DA used for that assay should be briefly mentioned/presented in the main text. The protocol for conducting this assay is not clear (page 9). Indeed, the density (of cells?) is not well presented: 1.5*106/mL. Does this mean 1.5*10E6 cells/mL? If not the case, the “adequate amount” of cells used should be provided for each cell line.
-Page 2, line 56, the authors mention values of ROS concentrations in healthy tissues and in cancer cells. One or more references should refer to these specific values.
-Page 3, lines 89-90, the use of the coumarin scaffold as latent fluorophore self-immolative core was already described for theranostic prodrug systems and some references should be cited: Chem. Commun. 2010, 46, 553−555 and Chem. Commun., 2017,53, 9470-9473.
-The paragraph on page 4 (line 101-116) should be proofread carefully because several sentences do not make sense (i.e. lines 103-104, lines 107-108, lines 114-115).
- Page 4, line 115, the sentence “The fluorescence intensities were dependent on the concentration of H2O2.” should be incremented by a comment on the kinetics since the increase of the fluorescence is correlated to the increase of the rate of the reaction (due to higher concentrations of H2O2).
-The organic synthesis experimental part should be also proofread carefully (e.g. p.8, line 241, the amount of bromohydroxybenzaldehyde is probably 1005 mg instead of 1005g!)
Author Response
Dear Editor:
This is the revision of molecules-2093278.
We sincerely thank you for your patience and careful work. We have addressed all the concerns and questions raised by the reviewers. Following is the point-by-point responses to all the comments.
Reviewers' comments:
Reviewer #1
This manuscript by Jiang et al. describes antitumor prodrugs of NAMPT inhibitors masked by a benzeneboronate moiety activatable by ROS in vitro. The technology relies on the boron-carbon bond oxidation by ROS such as hydrogen peroxide and further quinone methide self-immolation process for the release of the inhibitor FK866.
Overall, the chemistry and the biological results are sound and this study represents an interesting contribution. I would like to support the publication in Molecules provided that the authors answer my questions and remarks below.
- The comparative assay of ROS production by different cell lines led to the identification of PC-3 as the most productive cell line. The probe DCFH-DA used for that assay should be briefly mentioned/presented in the main text. The protocol for conducting this assay is not clear (page 9). Indeed, the density (of cells?) is not well presented: 1.5*106/mL. Does this mean 1.5*10E6 cells/mL? If not the case, the “adequate amount” of cells used should be provided for each cell line.
Response: Thank you for your useful suggestion. “DCFH-DA (2,7-Dichlorodihydrofluorescein diacetate) was used as an indicator for ROS in cells.” The description has been added in the revised manuscript.
Yes, this mean1.5*10E6 cells/mL and we have revised it in the manuscript.
-Page 2, line 56, the authors mention values of ROS concentrations in healthy tissues and in cancer cells. One or more references should refer to these specific values.
Response: Thank you for your careful work. References have been inserted.
-Page 3, lines 89-90, the use of the coumarin scaffold as latent fluorophore self-immolative core was already described for theranostic prodrug systems and some references should be cited: Chem. Commun. 2010, 46, 553−555 and Chem. Commun., 2017,53, 9470-9473.
Response: Thank you for your careful work. References have been inserted.
-The paragraph on page 4 (line 101-116) should be proofread carefully because several sentences do not make sense (i.e. lines 103-104, lines 107-108, lines 114-115).
Response: Thank you for your useful suggestion. The paragraph have been carefully revised.
“As shown in figure 3a, when compound 122-052 was treated with 100 μM H2O2, the oxidative intermediate 6 was appeared quickly and the active compound FK866 was also detected immediately. Subsequently, compound 6 further converted completely to FK866 with the extension of time. As shown in figure 3b, compound 122-066 was also treated with 100 μM H2O2. Similarly, the phenol intermediate 7 was observed firstly and then the released FK866 was appeared. However, the conversion process took a lot longer. Benefiting for the coumarin fluorophore, the chemical conversion of compound 122-066 under the treatment of H2O2 was also detected by fluorescence spectroscopy. The boronate moiety was oxidated by H2O2, and following a 1,6-elimination reaction, FK866 and the coumarin probe 8 were released. Probe 8 will then convert into compound 9 upon reaction with H2O”
- Page 4, line 115, the sentence “The fluorescence intensities were dependent on the concentration of H2O2.” should be incremented by a comment on the kinetics since the increase of the fluorescence is correlated to the increase of the rate of the reaction (due to higher concentrations of H2O2).
Response: Thank you for your useful suggestion. The following description was added in the revised manuscript. “When the concentration of H2O2 was increased, which would accelerate the rate of the reaction to release the fluorescent probe, exactly, the increase of the fluorescence was observed.”
-The organic synthesis experimental part should be also proofread carefully (e.g. p.8, line 241, the amount of bromohydroxybenzaldehyde is probably 1005 mg instead of 1005g!)
Response: Thank you for your careful work. We have proofread the manuscript carefully.
Reviewer #2
The authors developed new NAMPT inhibitors pro-drugs that become active after exposure to ROS starting from the FK866 backbone.
In this work, the authors show the synthesis process of the pro-drugs and some biological tests to confirm their specific activity in cells with high amounts of ROS. Compounds 122-052 and 122-066 showed higher cytotoxicity in cell lines with higher ROS levels.
This work aims to produce new NAMPT inhibitors that become active only in cancer cells with high ROS levels trying, in this way, to remove the known side effects of NAMPT inhibitors.
General concept comments
I found the work well done but the biological experiments are not enough to confirm, without any doubt, that these pro-drugs will be efficient in the whole organism and that they have no toxicity in megakaryocytes and lymphocytes as the native FK866.
How we can be sure that the pro-drugs will be stable in our body and will not become active drugs in some body's compartments? The authors should provide some pieces of evidence about the stability of the pro-dugs in some physiological conditions (e.g. liver metabolic activity)
Response: Thank you for your useful suggestion. The stability of the prodrugs were evaluated.
“The stability of the prodrugs 122-052 and 122-066 was evaluated in PBS (Phosphate Buffered Saline, pH 7.4) and liver microsome. The results showed that both of the compounds are very stable in the PBS and only a tiny compound degraded during the incubation for four days. In liver microsome, 122-052 still remained high stability, while 122-066 degraded along with the time (Figure 3). ”
The authors should test these pro-drugs on the cells that usually are affected by FK866: megakaryocytes and lymphocytes. I suggest a test on native and activated PBMCs, at least, to understand if the pro-drugs became active also in these cells.
Response: Thank you for your helpful suggestion. A test on native and activated PBMCs was performed.
“Cell viability test on native and activated PBMC (peripheral blood mononuclear cell) by phytohematoagglutinin (PHA, 5 µg/mL) was performed. The results showed that FK866 inhibited the cell viability remarkably on both native and activated PBMC at 4 nM. Whereas, cells were mostly unaffected by the treatment of prodrugs 122-052 and 122-066 at the same concentration (Figure 7).”
Specific comments
The legend of figure 1 should be more detailed. It should be explained the different parts of the molecules that have different colors.
Response: Thank you for your careful work. As your suggestion, the legend of figure 1 have been more detailed in the revised manuscript.
To highlight the changes that we have made, the “track changes'” versions of the manuscript have also been submitted.
We hope that the revised manuscript is acceptable for a speedy publication. Thank you.
Yours sincerely,
Hongli Chen

Reviewer 2 Report
Brief summary
The authors developed new NAMPT inhibitors pro-drugs that become active after exposure to ROS starting from the FK866 backbone.
In this work, the authors show the synthesis process of the pro-drugs and some biological tests to confirm their specific activity in cells with high amounts of ROS. Compounds 122-052 and 122-066 showed higher cytotoxicity in cell lines with higher ROS levels.
This work aims to produce new NAMPT inhibitors that become active only in cancer cells with high ROS levels trying, in this way, to remove the known side effects of NAMPT inhibitors.
General concept comments
I found the work well done but the biological experiments are not enough to confirm, without any doubt, that these pro-drugs will be efficient in the whole organism and that they have no toxicity in megakaryocytes and lymphocytes as the native FK866.
How we can be sure that the pro-drugs will be stable in our body and will not become active drugs in some body's compartments? The authors should provide some pieces of evidence about the stability of the pro-dugs in some physiological conditions (e.g. liver metabolic activity)
The authors should test these pro-drugs on the cells that usually are affected by FK866: megakaryocytes and lymphocytes. I suggest a test on native and activated PBMCs, at least, to understand if the pro-drugs became active also in these cells.
Specific comments
The legend of figure 1 should be more detailed. It should be explained the different parts of the molecules that have different colors.
Author Response
Dear Editor:
This is the revision of molecules-2093278.
We sincerely thank you for your patience and careful work. We have addressed all the concerns and questions raised by the reviewers. Following is the point-by-point responses to all the comments.
Reviewer #2
The authors developed new NAMPT inhibitors pro-drugs that become active after exposure to ROS starting from the FK866 backbone.
In this work, the authors show the synthesis process of the pro-drugs and some biological tests to confirm their specific activity in cells with high amounts of ROS. Compounds 122-052 and 122-066 showed higher cytotoxicity in cell lines with higher ROS levels.
This work aims to produce new NAMPT inhibitors that become active only in cancer cells with high ROS levels trying, in this way, to remove the known side effects of NAMPT inhibitors.
General concept comments
I found the work well done but the biological experiments are not enough to confirm, without any doubt, that these pro-drugs will be efficient in the whole organism and that they have no toxicity in megakaryocytes and lymphocytes as the native FK866.
How we can be sure that the pro-drugs will be stable in our body and will not become active drugs in some body's compartments? The authors should provide some pieces of evidence about the stability of the pro-dugs in some physiological conditions (e.g. liver metabolic activity)
Response: Thank you for your useful suggestion. The stability of the prodrugs were evaluated.
“The stability of the prodrugs 122-052 and 122-066 was evaluated in PBS (Phosphate Buffered Saline, pH 7.4) and liver microsome. The results showed that both of the compounds are very stable in the PBS and only a tiny compound degraded during the incubation for four days. In liver microsome, 122-052 still remained high stability, while 122-066 degraded along with the time (Figure 3). ”
The authors should test these pro-drugs on the cells that usually are affected by FK866: megakaryocytes and lymphocytes. I suggest a test on native and activated PBMCs, at least, to understand if the pro-drugs became active also in these cells.
Response: Thank you for your helpful suggestion. A test on native and activated PBMCs was performed.
“Cell viability test on native and activated PBMC (peripheral blood mononuclear cell) by phytohematoagglutinin (PHA, 5 µg/mL) was performed. The results showed that FK866 inhibited the cell viability remarkably on both native and activated PBMC at 4 nM. Whereas, cells were mostly unaffected by the treatment of prodrugs 122-052 and 122-066 at the same concentration (Figure 7).”
Specific comments
The legend of figure 1 should be more detailed. It should be explained the different parts of the molecules that have different colors.
Response: Thank you for your careful work. As your suggestion, the legend of figure 1 have been more detailed in the revised manuscript.
To highlight the changes that we have made, the “track changes'” versions of the manuscript have also been submitted.
We hope that the revised manuscript is acceptable for a speedy publication. Thank you.
Yours sincerely,
Hongli Chen

Round 2
Reviewer 2 Report
The answers given by the authors are exhaustive for me. I only suggest discussing in the "Conclusions" paragraph the non-toxicity of the compounds on PBMC regarding the previous literature about leucopenia and making some hypothesis why the compound 122-066 is degraded by liver microsome instead 122-052 is not degraded.
Author Response
Dear Editor:
This is the revision of molecules-2093278.
We sincerely thank you for your patience and careful work. We have addressed all the concerns and questions raised by the reviewers. Following is the point-by-point responses to all the comments.
Reviewers' comments:
The answers given by the authors are exhaustive for me. I only suggest discussing in the paragraph the non-toxicity of the compounds on PBMC regarding the previous literature about leucopenia and making some hypothesis why the compound 122-066 is degraded by liver microsome instead 122-052 is not degraded.
Response: Thank you for your useful suggestion. The following description has been added in the "Conclusions" pargraph.
“Meawhile, the toxicity of the prodrugs 122-052 and 122-066 was significantly reduced on both native and active PBMCs compared with FK866.”
“However, the coumarin moiety can cause the instability of prodrug 122-066 under rat liver microsomes, which is probably because coumarin can be metabolized by liver microsomes. [27, 28]”.
To highlight the changes that we have made, the “track changes'” versions of the manuscript have also been submitted.
We hope that the revised manuscript is acceptable for a speedy publication. Thank you.
Yours sincerely,
Hongli Chen